# Dietary Dihydroartemisinin Supplementation Attenuates Hepatic Oxidative Damage of Weaned Piglets with Intrauterine Growth Retardation through the Nrf2/ARE Signaling Pathway

**DOI:** 10.3390/ani9121144

**Published:** 2019-12-13

**Authors:** Yongwei Zhao, Yu Niu, Jintian He, Lili Zhang, Chao Wang, Tian Wang

**Affiliations:** College of Animal Science and Technology, Nanjing Agricultural University, No. 6, Tongwei Road, Xuanwu District, Nanjing 210095, China; zhaoyongweinjau@163.com (Y.Z.); niuyu0227@126.com (Y.N.); 15150537273@163.com (J.H.); zhanglili@njau.edu.cn (L.Z.); wangchao121@njau.edu.cn (C.W.)

**Keywords:** dihydroartemisinin, intrauterine growth retardation, liver, oxidative damage

## Abstract

**Simple Summary:**

Intrauterine growth retardation (IUGR) is usually defined as fetal growth below the tenth percentile for gestational age and results in impaired development and growth of the fetus during gestation. In addition to the high rates of perinatal mortality, IUGR has recently been shown to increase the risk of oxidative damage. Therefore, it is important to improve the body’s antioxidant capacity and reduce the oxidative damage caused by IUGR. The nuclear erythroid 2-related factor 2/ antioxidant response element (Nrf2/ARE) signaling pathway plays an important role in the defense against oxidative damage by increasing the activities of antioxidant enzymes. Dihydroartemisinin (DHA) is traditionally used to treat malaria. In addition, DHA has protective effects through increasing the activity of antioxidant enzymes and genes and the protein expression of Nrf2. Our results showed that dietary dihydroartemisinin supplementation improved antioxidant status in piglets with IUGR. Therefore, DHA can alleviate oxidative damage induced by IUGR in animals.

**Abstract:**

The object of present study was to evaluate the effects of dihydroartemisinin (DHA) supplementation on the hepatic antioxidant capacity in IUGR-affected weaned piglets. Eight piglets with normal birth weight (NBW) and sixteen IUGR-affected piglets were selected. Piglets were weaned at 21 days. NBW and IUGR groups were fed a basal diet and the ID group was fed the basal diet supplemented with 80 mg/kg DHA for 28 days. The result indicated that compared with NBW piglets, IUGR-affected piglets increased (*p* < 0.05) the concentration of malondialdehyde (MDA) and decreased (*p* < 0.05) the serum activities of total superoxide dismutase (T-SOD), catalase (CAT), and glutathione peroxidase (GSH-Px). In addition, IUGR-affected piglets showed increased (*p* < 0.05) hepatic concentrations of protein carbonyl (PC), 8-hydroxy-2’-deoxyguanosine (8-OHdG), and oxidized glutathione (GSSG), and an increased GSSG:GSH value. IUGR-affected piglets exhibited lower (*p* < 0.05) activities of GSH-Px, T-SOD, total antioxidant capacity (T-AOC), and the concentration of glutathione (GSH). DHA supplementation decreased (*p* < 0.05) the serum concentration of MDA and increased the serum activities of T-AOC, T-SOD, GSH-Px, and CAT. The ID group showed decreased (*p* < 0.05) concentrations of MDA, PC, 8-OHdG, and GSSG, and a decreased GSSG:GSH value in the liver. The hepatic activity of T-SOD and the concentration of GSH were increased (*p* < 0.05) in the liver of ID group. IUGR-affected piglets downregulated (*p* < 0.05) mRNA expression of nuclear erythroid 2-related factor 2 (Nrf2), heme oxygenase 1 (HO-1), and CAT. DHA supplementation increased (*p* < 0.05) mRNA expression of Nrf2, HO-1, GPx1, and CAT in the ID group. In addition, the protein expression of Nrf2 was downregulated (*p* < 0.05) in the liver of IUGR-affected piglets and DHA supplementation increased (*p* < 0.05) the protein content of Nrf2 and HO-1. In conclusion, DHA may be beneficial in alleviating oxidative damage induced by IUGR through the Nrf2/ARE signaling pathway in the liver.

## 1. Introduction

Intrauterine growth retardation (IUGR), which is usually defined as impaired growth and development of the mammalian fetus during gestation [1,2], is a major problem in human medicine. It has been reported that the 10th percentile is the cutoff point for IUGR which is a definition used by many institutions [3]. About 5–10% of human infants suffer from IUGR due to in vivo and in vitro causes, such as undernutrition or uterine dysfunction. This leads to high morbidity and mortality [4,5]. In addition to the high rates of perinatal mortality, IUGR has recently been shown to increase the risk of oxidative damage and reports also suggest that fetal hepatocytes in infants affected by IUGR might be subjected to oxidative damage, which reduces their ability to detoxify the liver [6,7] The liver is the metabolic center of the body and plays an important role in the absorption and metabolism of nutrients. Many studies have reported that IUGR can cause severe oxidative damage to the liver of piglets [8,9]. Aydan [10] demonstrated that IUGR can destroy the dynamic balance of the oxidation–antioxidant system in the liver, which can cause damage to the body. It is well known that nuclear erythroid 2-related factor 2 (Nrf2) is a redox-sensitive transcription factor, which will translocate from the cytoplasm to the nucleus, and bind to antioxidant response element (ARE) sequentially, when stimulated by an electrophile or oxidant [11] and it plays a vital role in alleviating oxidative damage by enhancing the activities of antioxidant enzymes [12,13]. Previous studies have demonstrated that IUGR can impair the Nrf2/ARE signaling pathway, which would decrease expression levels of antioxidant enzymes and antioxidant-related genes [14,15]. Therefore, evaluation of the Nrf2/ARE signaling pathway is an effective method to alleviate oxidative damage caused by IUGR.

Artemisinin, derived from the Chinese plant *Artemisia annua* is effective against both drug-resistant and cerebral malaria-causing strains of *Plasmodium falciparum* [16,17]. Other analogues of artemisinin, such as dihydroartemisinin (DHA), also exhibited excellent antimalarial activity and are therefore used in clinical treatment of malaria. DHA, prepared by reducing artemisinin with sodium borohydride, is the main metabolite of artemisinin drugs in vivo. DHA is traditionally used to treat malaria. However, in recent years, it has also been discovered that DHA plays an important role in anti-inflammation, immunoregulation, and anti-organizational fibrosis [18,19]. In addition, some studies have shown that DHA has protective effects against oxidative damage through various mechanisms in cancer pathogenesis, including increasing the expression levels of antioxidant-related enzymes, genes, and proteins [20]. Yang [21] found that DHA might alleviate pulmonary fibrosis and myofibroblast-like processes in alveolar epithelial cells in bleomycin-induced rats by reducing oxidative damage. These results indicated that DHA may reduce oxidative damage in vivo, thereby alleviating oxidative damage to the body.

However, as far as we know, the effects of DHA in weaned piglets is very limited. In this study, DHA was first applied to IUGR-affected weaned piglets. We hypothesized that dietary DHA supplementation plays an effective role on alleviating hepatic oxidative damage caused by IUGR. Therefore, the present study was conducted to survey whether DHA supplementation could improve the oxidative damage caused by IUGR in weaned piglets through the Nrf2/ARE signaling pathway.

## 2. Materials and Methods

### 2.1. Ethical Statement

The present experimental procedures were carried out according to the Institutional Animal Care and Use Committee of Nanjing Agricultural University (NJAU-CAST-2018-034).

### 2.2. Animals and Diet Design

The dihydroartemisinin was obtained from the Dasf Biotechnology Co., Ltd. (Nanjing, Jiangsu, China). The experimental piglets were selected from 10 litters (Duroc × (Landrace × Yorkshire)) of newborn piglets. These piglets were born from sows of similar weight (197.53 ± 1.68 kg) and parity (three or four births). All the sows were fed the same commercial diet based on the nutritional requirements stipulated by the National Research Council (NRC) (2012). One normal birth weight (NBW) piglet and two IUGR-affected piglets were selected in each litter. A piglet was defined as intrauterine growth-restricted when its birth weight was two standard deviations below the mean birth weight of the total population [22]. Pigs of NBW were selected according to the standard deviation range of the birth weight of the IUGR-affected pigs. Specific determination methods were based on previous studies in the laboratory, namely, the normal piglet weight was 1.56 ± 0.02 kg while the birth weight of IUGR-affected piglets was 0.99 ± 0.03 kg. All piglets naturally suck sows until they become weaned at 21 days. At weaning, the piglets were divided into three experimental groups: NBW (fed a basal diet), IUGR (fed a basal diet), ID (IUGR fed a basal diet + 80 mg/kg DHA). There were 10 piglets in each group, half male and half female. The concentration of DHA used in this study was determined through preliminary experiments. Table 1 shows the chemical composition of the diet, which was formulated to meet the nutritional requirements of the piglets according to NRC (2012). The piglets were fed ad libitum with water and feed.

### 2.3. Sample Collection

At 49 days of age, blood was collected by jugular vein puncture and serum samples were obtained from the blood by centrifugation at 3000 × *g* for 15 min at 4 °C. Eight weaned piglets weighing near to the average body weight of each pen were selected for euthanasia 12 h after the last meal. All the piglets were euthanized by exsanguination after electrical stunning. The liver samples from the left lobe were then removed immediately, rapidly frozen in liquid nitrogen, and stored at −80 °C for further analysis.

### 2.4. Assay of Serum Antioxidant Enzyme Activities

Serum samples were obtained from the blood by centrifugation at 3000 × *g* for 15 min at 4 °C. The concentration of malondialdehyde (MDA) and GSH and the activities of total superoxide dismutase (T-SOD) and glutathione peroxidase (GSH-Px) and the total antioxidant capacity (T-AOC) and catalase (CAT) in the serum were determined using the corresponding kits (Nanjing Jiancheng Institute of Bioengineering, Nanjing, Jiangsu, China). The samples were assayed in triplicate. The results were obtained by using a microplate reader and were displayed as U per milliliter for CAT, T-SOD, GSH-Px, and T-AOC, nmol per milliliter for MDA, and milligram per liter for GSH. U is the international unit of the enzyme, which is the amount of the enzyme that is required to convert 1 mmol of substrate in 1 min.

### 2.5. Determination of Hepatic Concentration of MDA, PC, and 8-OHdG

The liver samples were homogenized with 0.9% (wt/vol) ice-cold physiological saline for 10 s and the supernatant was obtained after centrifuging at 4000 g for 10 min at 4 °C. The concentration of protein carbonyl (PC) and MDA were measured using commercial assay kits (Nanjing Jiancheng Institute of Bioengineering, Nanjing, Jiangsu, China). The content of 8-OHdG in the liver was determined using an enzyme-linked immunosorbent assay (ELISA) kit from Shanghai YILI Biological Technology Co., Ltd. (Shanghai, China).

### 2.6. Determination of Hepatic Antioxidant Enzyme Activities

The liver samples were homogenized with 0.9% (wt/vol) ice-cold physiological saline for 10 s and the supernatant was obtained after centrifuging at 4000 g for 10 min at 4 °C. The activities of CAT, T-SOD, GSH-Px, and T-AOC and the concentrations of GSH and oxidized glutathione (GSSG) were measured using the corresponding kits (Nanjing Jiancheng Institute of Bioengineering, Nanjing, Jiangsu, China). The samples were assayed in triplicate. The amount of protein in the liver was measured using the bicinchoninic acid (BCA) protein assay kits (Nanjing Jiancheng Institute of Bioengineering, Nanjing, Jiangsu, China). The results were obtained by using a microplate reader and were displayed as U per milligram of protein for CAT, T-SOD, GSH-Px, and T-AOC and µmol per gram of protein for GSH and GSSG. U is the international unit of the enzyme, which is the amount of the enzyme that is required to convert 1 mmol of substrate in 1 min.

### 2.7. Assay of Gene Expression

Total RNA was isolated from liver samples using TRIzol Reagent (TaKaRa Biotechnology, Dalian, Liaoning, China). RNA integrity was assessed on 1% agarose gels with ethidium bromide staining. The concentration and purity of the total RNA were assessed from OD 260/280 readings (ratio > 1.8) using a spectrophotometer (NanoDrop Technologies, Wilmington, DE, USA). Total RNA (1 µg) was reverse-transcribed into cDNA using the PrimeScript RT Reagent Kit (TaKaRa Biotechnology) according to the manufacturer’s guidelines. Quantitative real-time polymerase chain reaction (qRT-PCR) was performed on an ABI StepOnePlus Real-Time PCR System (Applied Biosystems, Grand Island, NY, USA) according to the manufacturer’s instructions. The sequences of primers used in this experiment are shown in Table 2. The cDNA samples were amplified by qRT-PCR with SYBR Premix Ex Taq reagents (Takara Biotechnology). Briefly, a 20 µL reaction mixture was prepared using 2 µL of cDNA, 0.4 µL each of forward and reverse primers, 0.4 µL of ROX reference dye (50×; Life Technologies, Grand Island, New York, USA), 10 µL of SYBR Premix Ex Taq (2×), and 6.8 µL of double-distilled H_2_O. Each sample was tested in duplicate. The qRT-PCR conditions consisted of an initial denaturation at 95 °C for 30 s, followed by 40 cycles of denaturation at 95 °C for 5 s and annealing at 60 °C for 30 s. The conditions of the melting curve analysis were one cycle of denaturation at 95 °C for 10 s, followed by an increase in temperature from 65 to 95 °C at a rate of 0.5 °C/s. The relative mRNA expression levels were calculated using the 2-ΔΔCT method, with glyceraldehyde-3-phosphate dehydrogenase (GAPDH) as the internal standard and the values for pigs in the NBW group were used as a calibrator.

### 2.8. Western Blotting

The primary antibodies against Nrf2 (dilution 1:600), HO-1 (dilution 1:600), and β-actin (dilution 1:4000) were obtained from Proteintech Group Inc. (Rosemont, IL, USA). Proteins were extracted from about 50 mg of the liver using radioimmunoprecipitation assay lysis buffer purchased from Beyotime Institute of Biotechnology (Nantong, Jiangsu, China). The protein concentrations of each sample were determined using a BCA Protein Assay Kit. About 60 µg proteins from each sample were electrophoresed in sodium dodecyl sulfate-polyacrylamidegel electrophoresis (SDS-PAGE) and transferred to polyvinylidene difluoride membranes. At room temperature, the membranes were sealed with sealing buffer for 2 h. The membranes were then washed four times and probed with the primary and secondary antibodies (Proteintech Group Inc.; horseradish-peroxidase-conjugated goat anti-rabbit Ig G; 1:5000). The blots were developed using enhanced chemiluminescence reagents (Beyotime Institute of Biotechnology, Nantong, Jiangsu, China) followed by autoradiography. Images of the membranes were recorded with the Luminescent Image Analyzer LAS-4000 system (Fujifilm, Tokyo, Japan) and quantified by the Gel-Pro Analyzer 4.0 software (Media Cybernetics, Silver Spring, MD, USA).

### 2.9. Statistical Analysis

SPSS 20.0 statistical software was used for data analysis. One-way analysis of variance and the Duncan method were used for statistical differences between different groups. *p* values less than 0.05 were considered as statistically significant. Data were are expressed as mean ± standard error values.

## 3. Results

### 3.1. Growth Performance 

The IUGR and ID groups had lower body weight at 21 days of age (p < 0.05) than the NBW group. At 49 days of age, the body weight of the NBW and ID groups did not differ (p > 0.05) and were greater than that of the IUGR group (p < 0.05). The data are provided in the Appendix A (Appendix A).

### 3.2. Serum Antioxidant Enzyme Activities

Compared with the NBW piglets, the IUGR-affected piglets exhibited higher concentrations (*p* < 0.05) of MDA (Table 3). The activities of CAT, T-SOD, and GSH-Px and were decreased (*p* < 0.05) in IUGR-affected piglets. In the ID group, the concentration of MDA was lower (*p* < 0.05) and the activities of T-AOC, T-SOD, GSH-Px, and CAT were increased (*p* < 0.05) compared with those in IUGR group.

### 3.3. Hepatic Antioxidant Enzyme Activities

Compared with the NBW piglets, the IUGR-affected piglets showed lower (*p* < 0.05) concentrations of GSH and decreased activities of T-SOD, T-AOC, and GSH-Px (Table 4). The concentration of GSSG and the GSSG:GSH value were increased in the IUGR group. Additionally, hepatic CAT activity was not affected by IUGR. In the ID group, T-SOD activity and the concentrations of GSH were higher (*p* < 0.05), the concentrations of GSSG and the GSSG:GSH value were lower (*p* < 0.05) compared with those in the IUGR group. Additionally, the activity of T-AOC and CAT in the ID group showed no difference compared with the IUGR group.

### 3.4. Hepatic Concentrations of MDA, PC and 8-OHdG

As shown in Figure 1, IUGR-affected piglets exhibited higher (*p* < 0.05) concentrations of PC and 8-OHdG in comparison with the NBW piglets. The hepatic levels of PC, MDA, and 8-OHdG were significantly lower (*p* < 0.05) in the ID piglets than those of IUGR-affected piglets.

### 3.5. Hepatic Antioxidant-Related Gene Expression

As shown in Figure 2, IUGR-affected piglets had downregulated (*p* < 0.05) mRNA expression levels of HO-1, Nrf2, and CAT compared with NBW piglets. After dietary supplementation with DHA, mRNA expression levels of HO-1, Nrf2, CAT, and GPx1 were upregulated (*p* < 0.05) compared with those in the IUGR group. Meanwhile, mRNA expression levels of Nrf2 and GPx1 in ID piglets were higher (*p* < 0.05) compared with those in NBW piglets.

### 3.6. Hepatic Protein Expression

As shown in Figure 3 and Figure 4, in the IUGR group, the protein expression of Nrf2 decreased (*p* < 0.05) and the protein expression of HO-1 showed a tendency to be downregulated compared with those in NBW group. Dietary DHA supplementation significantly increased (*p* < 0.05) the protein expression of Nrf2 and HO-1 in the ID group.

## 4. Discussion

According to statistics, in 2015, the number of children with stunted growth worldwide was 113.4 million [23]. In addition, there is more and more evidence that IUGR causes oxidative stress in offspring, which is evidenced by oxidative damage [24]. A previous study has found that IUGR attenuates the antioxidant capacity of various organs and causes oxidative damage in the body [10]. Posner [25] has demonstrated that artemisinin plays an important role in reducing oxidant damage in cancer pathogenesis. Moreover, it has been reported that artemisinin can improve the effectiveness of antioxidants on myocardial ischemia-reperfusion injury in rats [26]. Therefore, the present study was conducted to investigate the hepatic antioxidant capacity of the IUGR-affected piglets and whether DHA treatment could alleviate the oxidative damage caused by IUGR during the post-weaning period.

Oxidative damage is a pathophysiological response associated with many different diseases and it appears in patients with IUGR [27]. MDA is a marker of lipid peroxidation [28], the concentration of which directly reflect the degree of lipid peroxidation. In addition, PC is a marker of protein oxidative damage and 8-OHdG is used as a marker of DNA oxidation. In IUGR-affected piglets of this study, the hepatic concentrations of 8-OHdG and PC were significantly increased and the content of MDA in the serum was increased compared to NBW piglets, which was in accordance with the findings of Zhang [9]. In addition, SOD, GSH-Px, and CAT are important enzymes for scavenging free radicals in the body, which constitute the first line of the body’s antioxidant defense system. CAT is an endogenous antioxidant enzyme that detoxifies hydrogen peroxide to water and functions in the same pathway as SOD, which converts superoxide to hydrogen peroxide. In addition, SOD is essential for preventing oxidative damage in the body and is the first line of defense for antioxidant defense systems. In the present study, we observed that the hepatic activities of T-SOD, T-AOC, and GSH-px were significantly decreased and the activities of T-SOD, GSH-Px, and CAT were decreased in the serum of IUGR-affected piglets. GSH is a major component of the cellular antioxidant system, which can eliminate lipid peroxides and repair oxidative proteins through a reaction catalyzed by GSH-Px. During these reactions, GSH is converted to its disulfide form, GSSG. [29]. We found that the IUGR group exhibited lower levels of GSH and higher levels of GSSG and an increased GSSG:GSH value in the liver. These results demonstrated that in piglets, IUGR aggravated the degree of oxidative damage in the liver and reduced the body’s antioxidant capacity. These findings were similar to those reported by Jintian [15], who pointed out that IUGR caused severe damage to the liver’s antioxidant function in rats. DHA is used commonly as an anti-malarial agent in clinical treatment and some studies have indicated that it also plays a vital role in antioxidant system [25]. However, studies on the effects of DHA on oxidative damage in IUGR-affected piglets are very limited. In the present study, our results indicated that DHA administration decreased the concentrations of MDA and increased the activities of T-SOD, T-AOC, CAT, and GSH-Px in the serum. What is more, the concentrations of MDA, H_2_O_2_, PC, 8-OHdG, and GSSG and the GSSG:GSH value were decreased and the T-SOD activity and the concentrations of GSH were increased in the liver. Yang [21] reported that dietary DHA supplementation significantly increased the lung SOD activity and the concentrations of GSH in rats, which is consistent with our results.

In addition, to explore the exact mechanism of oxidative damage caused by IUGR, we detected the expression levels of genes and proteins related to hepatic antioxidation. Nrf2 is a key transcription factor in the body’s antioxidant defense system. When exposed to oxidative stressors, Nrf2 enters the nucleus and bound to the ARE, which activates a series of downstream phase II detoxification enzymes and antioxidant enzymes [30]. Nrf2 and its target genes HO-1 can improve the antioxidant capacity of cells and alleviate oxidative damage. In the present study, the hepatic mRNA expression of Nrf2 and HO-1 were downregulated in IUGR-affected piglets. At the same time, the mRNA expression of CAT was also significantly downregulated. But the mRNA expression of SOD1 and GPx1 showed no significant differences compared with NBW piglets. Similarly, Feng [8] found that there is no difference in the mRNA expression of GPx1 and SOD1 in low body weight piglets and NBW piglets. Diets supplemented with DHA significantly increased the hepatic mRNA expression of HO-1 and Nrf2 in IUGR-affected piglets. The mRNA expression levels of antioxidant-related genes involved in the Nrf2/ARE signaling pathway (CAT and GPx1) were significantly increased by DHA supplementation in IUGR-affected piglets. In addition, the hepatic protein expression of Nrf2 was downregulated and HO-1 exhibited a tendency to be downregulated in IUGR-affected piglets. DHA treatment significantly upregulated the hepatic protein expression of Nrf2 and HO-1 in IUGR-affected piglets, which is similar to a previous finding [21]. These results may indicate that DHA upregulates the Nrf2/ARE signaling pathway to prevent oxidative damage due to IUGR.

## 5. Conclusions

In conclusion, our results indicate that IUGR can cause severe oxidative damage to the liver of weaned piglets. Diets supplemented with 80 mg/kg DHA could efficiently enhance hepatic antioxidation capacity through the Nrf2/ARE signal pathway. However, further and more comprehensive studies are required to confirm this. Our research may aid in finding new strategies to treat IUGR in both animals and humans.

## Figures and Tables

**Figure 1 animals-09-01144-f001:**
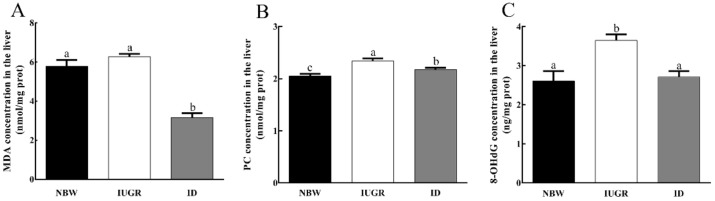
Effects of dietary dihydroartemisinin supplementation on the hepatic concentrations of MDA (**A**), PC (**B**), and 8-OHdG (**C**) in intrauterine growth retardation-affected weaned piglets. ^a,b,c^ Bars in each panel without a common superscript letter were significantly different (*p* < 0.05). Values were means and standard errors (*n* = 8). NBW, normal birth weight group given a control diet; IUGR, intrauterine growth retardation group given a control diet; ID, IUGR group given diets supplemented with 80 mg/kg DHA; PC, protein carbonyl; MDA, malondialdehyde; and 8-OHdG, 8-hydroxy-2’-deoxyguanosine.

**Figure 2 animals-09-01144-f002:**
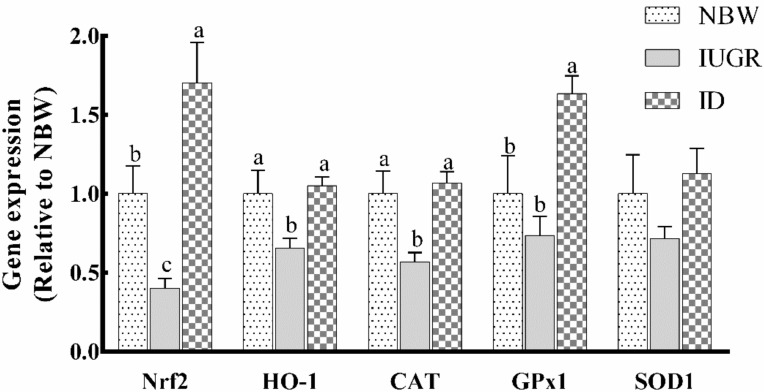
Effects of dietary DHA supplementation on the antioxidant related mRNA expressions in the liver of intrauterine growth retardation-affected weaned piglets. ^a,b,c^ Bars for each gene without a common superscript letter were significantly different (*p* < 0.05). Values were means and standard errors (*n* = 8). NBW, normal birth weight group given a control diet; IUGR, intrauterine growth retardation group given a control diet; ID, IUGR group given diets supplemented with 80 mg/kg DHA; CAT, catalase; SOD1, superoxide dismutase 1; GPx1, glutathione peroxidase 1; Nrf2, nuclear erythroid 2-related factor 2; and HO-1, heme oxygenase 1.

**Figure 3 animals-09-01144-f003:**
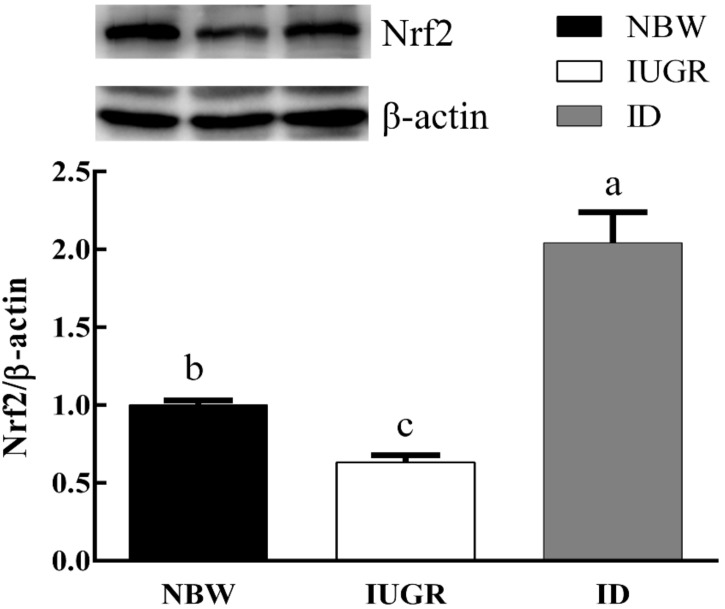
Effects of DHA supplementation on hepatic Nrf2 protein contents in intrauterine growth retardation-affected weaned piglets. ^a,b,c^ Bars in each panel without a common superscript letter were significantly different (*p* < 0.05). Values were means and standard errors (*n* = 8). NBW, normal birth weight group given a control diet; IUGR, intrauterine growth retardation group given a control diet; and ID, IUGR group given diets supplemented with 80 mg/kg DHA.

**Figure 4 animals-09-01144-f004:**
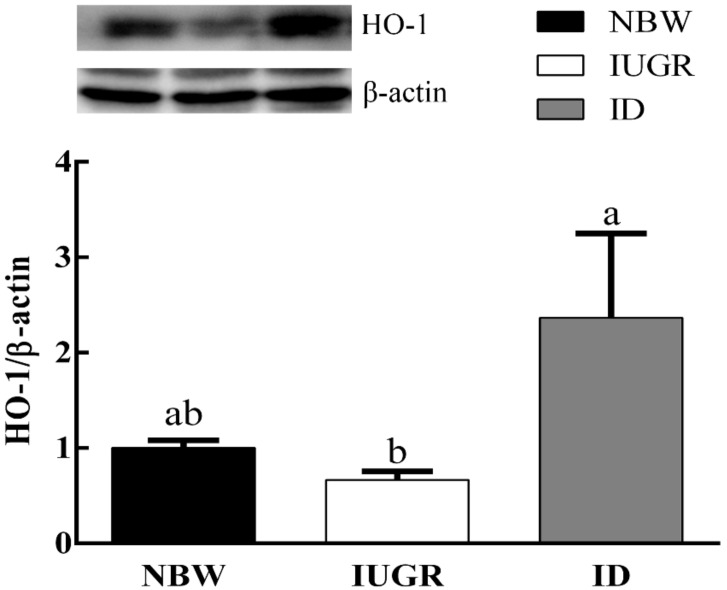
Effects of DHA supplementation on HO-1 protein contents in the liver of intrauterine growth retardation-affected weaned piglets. ^a,b,c^ Bars in each panel without a common superscript letter were significantly different (*p* < 0.05). Values were means and standard errors (*n* = 8). NBW, normal birth weight group given a control diet; IUGR, intrauterine growth retardation group given a control diet; and ID, IUGR group given diets supplemented with 80 mg/kg DHA.

**Table 1 animals-09-01144-t001:** Compositions of the basal diets (as-fed basis).

Items	Content (%)
**Ingredients (%)**	
Corn	65.00
Soybean meal	10.00
Fish meal	4.00
Extruded soybean	8.00
Whey power	5.00
Fermented soybean meal	4.00
Premix ^1^	4.00
Total	100.00
**Nutrient Level**	
Crude protein (%)	18.15
Gross energy (MJ/kg)	9.00
Digestible energy (MJ/kg)	14.58
Metabolisable energy (MJ/kg)	11.41
Lysine (%)	1.30
Methionine (%)	0.32
Methionine + Cystine (%)	0.60
Threonine (%)	0.83
Ca (%)	0.71
Total phosphorus (%)	0.72
Available phosphorus (%)	0.27

^1^ In premix, provided per mg/kg of diet: cholecalciferol, 0.075; retinyl acetate, 4.79; menadione, 3; all-rac-α-tocopherol acetate, 100; riboflavin, 8; thiamin, 3; cobalamin, 0.04; nicotinamide, 5; pantothenic acid, 20; niacin, 45; folic acid, 2; biotin, 0.3; choline chloride, 450; Fe, 180; Cu, 230; Zn (as ZnO), 65; Mn, 50; I, 0.5; and Se, 0.2. Dihydroartemisinin diets: basic diets + 80 mg/kg dihydroartemisinin.

**Table 2 animals-09-01144-t002:** Sequences for real-time PCR primers.

Gene ^1^	Accession Number	Sequences (5’–3’)	Product Length (bp)
GAPDH	NM_001206359.1	F:TCGGAGTGAACGGATTTGGC	189
		R: TGACAAGCTTCCCGTTCTCC	
Nrf2	XM_005671981.3	F:GACTCAAGGGGTTGCGAAGG	80
		R: CCCAAACCCCAATCCCGTAG	
CAT	NM_214301.2	F: CCTGCAACGTTCTGTAAGGC	109
		R: ATATCAGGTTTCTGCGCGGC	
SOD1	NM_001190422.1	F:TGAAGGGAGAGAAGACAGTGTTA	130
		R: GGATTGAAGTGAGGACCTGC	
GPx1	NM_214201.1	F: CTCATGACCGACCCCAAGTT	128
		R: GTCAGAAAGCGACGGCTGTA	
HO-1	NM_001004027.1	F: TGTACCGCTCCCGAATGAAC	142
		R: TGGTCCTTAGTGTCCTGGGT	

^1^ GAPDH, glyceraldehyde-3-phosphate dehydrogenase; GPx1, glutathione peroxidase 1; Nrf2, nuclear erythroid 2-related factor 2; SOD1, superoxide dismutase 1; CAT, catalase; and HO-1, heme oxygenase 1.

**Table 3 animals-09-01144-t003:** Effects of dietary dihydroartemisinin supplementation on the serum redox status of intrauterine growth retardation-affected weaned piglets.

Item ^1^	Experiment Groups		
	NBW	IUGR	ID
T-AOC (U/mL)	1.316 ± 0.076 ^a,b^	1.233 ± 0.090 ^b^	1.604 ± 0.123 ^a^
T-SOD (U/mL)	220.235 ± 11.695 ^a^	176.722 ± 7.423 ^b^	211.167 ± 4.501 ^a^
GSH-Px (U/mL)	294.339 ± 3.551 ^a^	252.579 ± 3.345 ^b^	287.799 ± 5.507 ^a^
GSH (mg/L)	8.725 ± 1.143	6.363 ± 1.028	5.838 ± 0.834
CAT (U/mL)	6.835 ± 0.592 ^a^	3.674 ± 0.367 ^b^	6.248 ± 0.445 ^a^
MDA (nmol/mL)	8.101 ± 0.578 ^b^	13.445 ± 0.832 ^a^	7.952 ± 0.463 ^b^

^a,b^ Means in a row without a common superscript letter were significantly different (*p* < 0.05). Values were means and standard errors (*n* = 8). ^1^ NBW, normal birth weight group given a basal diet; IUGR, intrauterine growth retardation group given a basal diet; ID, IUGR group given diets supplemented with 80 mg/kg dihydroartemisinin; T-AOC, total antioxidant capacity; T-SOD, total superoxide dismutase; GSH-Px, glutathione peroxidase; and CAT, catalase.

**Table 4 animals-09-01144-t004:** Effects of dietary dihydroartemisinin supplementation on the hepatic redox status of intrauterine growth retardation-affected weaned piglets.

Item ^1^	Experiment Groups		
	NBW	IUGR	ID
CAT (U/mg protein)	12.856 ± 0.623	11.481 ± 0.196	12.091 ± 0.422
T-AOC (U/mg protein)	1.207 ± 0.110 ^a^	0.800 ± 0.046 ^b^	1.009 ± 0.076 ^a,b^
T-SOD (U/mg protein)	200.104 ± 8.91 ^a^	154.893 ± 4.273 ^b^	202.152 ± 7.650 ^a^
GSH-Px (U/mg protein)	154.469 ± 7.100 ^a^	125.428 ± 3.605 ^b^	133.493 ± 5.228 ^b^
GSH (µmol/g protein)	24.945 ± 1.669 ^a^	17.152 ± 0.817 ^c^	20.880 ± 0.706 ^b^
GSSG (µmol/g protein)	31.26 ± 0.371 ^b^	39.755 ± 0.404 ^a^	31.998 ± 0.406 ^b^
GSSG:GSH	1.355 ± 0.147 ^b^	2.191 ± 0.144 ^a^	1.658 ± 0.101 ^b^

^a,b,c^ Means in a row without a common superscript letter were significantly different (*p* < 0.05). Values were means and standard errors (*n* = 8). ^1^ NBW, normal birth weight group given a control diet; IUGR, intrauterine growth retardation group given a control diet; ID, IUGR group given diets supplemented with 80 mg/kg; T-AOC, total antioxidant capacity; T-SOD, total superoxide dismutase; GSH-Px, glutathione peroxidase; CAT, catalase; and GSSG, oxidized glutathione.

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
