# Peer review of "Dietary Dihydroartemisinin Supplementation Attenuates Hepatic Oxidative Damage of Weaned Piglets with Intrauterine Growth Retardation through the Nrf2/ARE Signaling Pathway"

_animals, 2019, doi:10.3390/ani9121144_

Round 1

Reviewer 1 Report

If possible, could the authors explain the concentration of DHA used in the study? Why this concentration? 

This is not clear for the reader.

Author Response

Point 1: If possible, could the authors explain the concentration of DHA used in the study? Why this concentration? This is not clear for the reader.

Response: Thank you very much for your suggestion. DHA was applied to weaned piglets for the first time in this experiment. Before this experiment, we conducted a DHA concentration gradient test to choice the best concentration of DHA added to weaned piglets. According to the previous studies on mice [1-4], we designed five groups, namely the control group (without DHA), 20 mg/kg DHA group, 40 mg/kg DHA group, 80 mg/kg DHA group and 160 mg/kg DHA group. We selected 21-day-old weaned piglets for a test period of 30 days. On the 50th day, we measured the growth performance and serum biochemical indicators. It was found that 80 mg/kg DHA significantly improved the growth performance of weaned piglets, and the serum biochemical indexes and blood routine indexes were also better than other doses. Therefore, under the condition of this experiment, the best concentration of DHA in weaned piglets is 80 mg/kg, so this experiment chose this dosage for follow-up study. As the relevant data has been delivered to other Journals, it cannot be listed in this manuscript to avoid data reuse.

References:

Yang D X , Qiu J , Zhou H H , et al. Dihydroartemisinin alleviates oxidative stress in bleomycin-induced pulmonary fibrosis[J]. Life Sciences, 2018:S0024320518302790. Wei M , Xie X , Chu X , et al. Dihydroartemisinin suppresses ovalbumin-induced airway inflammation in a mouse allergic asthma model[J]. Immunopharmacology and Immunotoxicology, 2013, 35(3):382-389. Posobiec L M , Clark R L , Bushdid P B , et al. Dihydroartemisinin (DHA) Treatment Causes an Arrest of Cell Division and Apoptosis in Rat Embryonic Erythroblasts in Whole Embryo Culture[J]. Birth Defects Research Part B: Developmental and Reproductive Toxicology, 2013, 98(6):445-458. Xie L H , Li Q , Zhang J , et al. Pharmacokinetics, tissue distribution and mass balance of radiolabeled dihydroartemisinin in male rats[J]. Malaria Journal, 2009, 8(1):112-0.

Reviewer 2 Report

The authors show that intrauterine growth retardation (IUGR) results in decreased expression and activity of antioxidant enzymes and increased oxidative damage in piglets and that supplementation with 80 mg/kg dihydroartemisinin (DHA) for 4 weeks starting at weaning upregulates the Nrf2 antioxidant pathway to reverse many of these changes.  The methodologies are not well-explained, but overall the manuscript is well-organized and the data supports the conclusions. The English grammar could also be improved.

Major comments:

Comment 1: Was the weight of the piglets taken after the 4 weeks of treatment before euthanasia? If so, id DHA affect the rate of weight gain?

Comment 2: (optional) The method of indicating statistical significance in the tables could be simplified. When a value is statistically different from all other values in that same row, no superscript could be given. For example in the middle data column (IUGR) in Table 3, the superscript could be removed in rows 2,3,5, and 6 as these values are significantly different from the other two.

Comment 3: Line 118: The methods for measuring malondialdehyde (MDA) and GSH and the activities of total superoxide dismutase (T-SOD), glutathione peroxidase (GSH-Px), total antioxidant capacity (T-AOC) and catalase (CAT) in the serum should be described in more detail. Measuring the GSSG:GSH ratio is more informative of antioxidant potential than measures of GSH by itself.

Comment 4: Line 130: The methods for measuring CAT, T-SOD, GSH-Px, T-AOC and the concentrations of GSH and H2O2 should be described in more detail. Methods of measuring H2O2 are usually not accurate after freezing tissue due to freeze-thaw damage of mitochondrial membranes where most superoxide is produced and then converted to superoxide by SODs.

Line 264: hydroxyl radicals -> hydrogen peroxide

Line 265: T-AOC -> SOD

Minor Comments:

Wording changes:

Line 3 (title): attenuates on -> attenuates

Line 99: can naturally -> naturally

Line 100: within 21 days of weaning -> until they become weaned at 21 days

Line 179: T-AOC、T-SOD、GSH-Px -> T-AOC, T-SOD, GSH-Px,

Lines 209, 235, and 242: Means in a row -> Bars in each panel

Line 217: GSH-Px -> GPx1

Line 222: Means in a row -> Bars for each gene

Line 228: IUGR -> the IUGR

Lines 235 and 242: Means in a row -> Bars

Line 247: million. -> million

Line 248: evidenced -> is evidenced

Line 248: damages -> damage

Line 248: The previous -> A previous

Line 250: had -> has

Line 250: played -> plays

Line 257: ,the -> , the

Line 262: substances -> enzymes

Line 264: removes hydroxyl radicals -> detoxifies hydrogen peroxide to water

Line 264: synergizes -> functions in the same pathway

Line 264&265: SOD to convert superoxide anions into water -> SOD, which converts superoxide to hydrogen peroxide

Line 266: Remove sentence (redundant) “The vitality of T-AOC can reflect endogenous antioxidant capacity.”

Line 267: present -> the present

Line 268: T-SOD 、-> T-SOD,

Line 269: was -> is

Line 271: was -> is

Line 272: signal -> signaling

Line 274: damagr -> damage

Line 277: was used usually -> is used commonly

Line 281: T-SOD、T-AOC、CAT -> T-SOD, T-AOC, CAT,

Line 284: was -> is

Line 302: repair the oxidative damage caused by IUGR through the Nrf2/ARE signaling pathway ->

                 upregulates the Nrf2/ARE signaling pathway to prevent oxidative damage due to IUGR.

Line 305: indicated -> indicate

Line 307: through -> through the

Author Response

Response to Reviewer 2 Comments

Major comments:

Point 1: Was the weight of the piglets taken after the 4 weeks of treatment before euthanasia? If so, id DHA affect the rate of weight gain?

Response: Thank you very much for your suggestion. We measured the body weights of normal and IUGR at both weaning and final sampling age. At 21d, NBW: 6.96 kg±0.05. IUGR: 6.08 kg ±0.09, IUGR-DHA: 6.08 kg ±0.06. At 49d, NBW: 12.80 kg ±0.60, IUGR: 10.24 kg ±0.38, IUGR-DHA: 12.11 kg ±0.46. DHA can significantly increase weight of the IUGR piglets. These data have been published in other journals, so it is not described in this manuscript.

Point 2: (optional) The method of indicating statistical significance in the tables could be simplified. When a value is statistically different from all other values in that same row, no superscript could be given. For example in the middle data column (IUGR) in Table 3, the superscript could be removed in rows 2,3,5, and 6 as these values are significantly different from the other two.

Response: Thank you very much for your suggestion. Your suggestion is very meaningful. However, the size of the data does not reflect whether the difference is significant. The superscript letters can make the readers more clear about the data. In addition, the superscript letters make the full text unified, which is more conducive to readers' understanding. If you have any questions, please don't hesitate to contact us.

Point 3: Line 118: The methods for measuring malondialdehyde (MDA) and GSH and the activities of total superoxide dismutase (T-SOD), glutathione peroxidase (GSH-Px), total antioxidant capacity (T-AOC) and catalase (CAT) in the serum should be described in more detail. Measuring the GSSG:GSH ratio is more informative of antioxidant potential than measures of GSH by itself.

Response: Thank you very much for your suggestion. Your suggestion is very meaningful. We have described the methods for these indicators in more detail. And we have added the concentration of GSSG and the GSSG:GSH ratio in the Table 4. If you have any questions, please don't hesitate to contact us.

Point 4: Line 130: The methods for measuring CAT, T-SOD, GSH-Px, T-AOC and the concentrations of GSH and H2O2 should be described in more detail. Methods of measuring H2O2 are usually not accurate after freezing tissue due to freeze-thaw damage of mitochondrial membranes where most superoxide is produced and then converted to superoxide by SODs.

Response: Thank you very much for your suggestion. Your suggestion is very important for improving the quality of our manuscripts. We have described the methods for these indicators in more detail. And It is our fault. The method of measuring H2O2 is indeed inaccurate. Considering your suggestion and the scientific nature of the article, we have deleted this indicator. If you have any questions, please don't hesitate to contact us.

Point 5: Line 264: hydroxyl radicals -> hydrogen peroxide

Response: Thanks for your comments. We have corrected it in our manuscript.

Point 6: Line 265: T-AOC -> SOD

Response: Thanks for your comments. We have corrected it in our manuscript.

Minor Comments:

Wording changes:

Line 3 (title): attenuates on -> attenuates

Line 99: can naturally -> naturally

Line 100: within 21 days of weaning -> until they become weaned at 21 days

Line 179: T-AOC、T-SOD、GSH-Px -> T-AOC, T-SOD, GSH-Px,

Lines 209, 235, and 242: Means in a row -> Bars in each panel

Line 217: GSH-Px -> GPx1

Line 222: Means in a row -> Bars for each gene

Line 228: IUGR -> the IUGR

Lines 235 and 242: Means in a row -> Bars

Line 247: million. -> million

Line 248: evidenced -> is evidenced

Line 248: damages -> damage

Line 248: The previous -> A previous

Line 250: had -> has

Line 250: played -> plays

Line 257: ,the -> , the

Line 262: substances -> enzymes

Line 264: removes hydroxyl radicals -> detoxifies hydrogen peroxide to water

Line 264: synergizes -> functions in the same pathway

Line 264&265: SOD to convert superoxide anions into water -> SOD, which converts superoxide to hydrogen peroxide

Line 266: Remove sentence (redundant) “The vitality of T-AOC can reflect endogenous antioxidant capacity.”

Line 267: present -> the present

Line 268: T-SOD 、-> T-SOD,

Line 269: was -> is

Line 271: was -> is

Line 272: signal -> signaling

Line 274: damagr -> damage

Line 277: was used usually -> is used commonly

Line 281: T-SOD、T-AOC、CAT -> T-SOD, T-AOC, CAT,

Line 284: was -> is

Line 302: repair the oxidative damage caused by IUGR through the Nrf2/ARE signaling pathway ->upregulates the Nrf2/ARE signaling pathway to prevent oxidative damage due to IUGR.

Line 305: indicated -> indicate

Line 307: through -> through the

Response: Thanks for your comments. We are very sorry for our writing mistakes. We have corrected these words in our revised manuscript.

Reviewer 3 Report

The authors examined the effects of dietary inclusion of dihydroartemisinin on the hepatic oxidative damage in weaning piglets.

The manuscript was well written and l would acknowledge that after minor language editing the manuscript can be accepted for publication.

Author Response

Response to Reviewer 3 Comments

Point 1: The manuscript was well written and l would acknowledge that after minor language editing the manuscript can be accepted for publication.

Response: Thanks for your comments. We have edited the language of the manuscript. If you have any questions, please don't hesitate to contact us.
